# Determinants and constraints of feather growth

**Lukas Jenni**[1]*, **Kathrin Ganz**[1], **Pietro Milanesi**[1], **Raffael Winkler**[2]

**1** Swiss Ornithological Institute, Sempach, Switzerland, **2** Natural History Museum of Basel, Basel, Switzerland

* lukas.jenni@vogelwarte.ch

**Data Availability Statement:** Data are available in the Zenodo repository at https://doi.org/10.5281/zenodo.3724459.

**Funding:** The author(s) received no specific funding for this work.

## Abstract

During the periodic moult of the plumage of birds, a fast regrowth of feathers would shorten the time of reduced plumage functionality. However, it has long been known that feather growth-rate is limited and that long feathers take disproportionally longer to grow than small feathers, which has severe consequences on moult duration and the completeness of moult in large birds. The reasons for the limitations of feather-growth must be related to the size and/or functions of the feather follicle, but are largely unknown. Here we measured the size of the feather follicle (taking calamus width as a proxy) and related it to parameters of feather growth (feather growth-rate by mass and by length) and feather structure (feather length, mass, massiveness [mass of feather material per mm feather-length]). We used three independent datasets which allowed for interspecific analyses, and for intraspecific comparisons of differently structured feathers within the framework of biological scaling. We found that the cross-sectional area of the calamus (as a proxy of feather follicle size) was directly proportional to feather growth-rate by mass. Hence, factors acting at a two-dimensional scale (possibly nutrient supply to the growing feather) determines feather growth rate by mass, rather than the linear arrangement of stem cells (in a circular configuration) as had previously been assumed. Feather follicle size was correlated with both feather length and massiveness, hence it seems to be adapted to some extent to feather structure. Feather growth-rate by length was dependent on both the feather material produced per unit time (growth-rate by mass) and the amount of material deposited per unit feather-length. Follicle size not only determines feather growth-rate by mass, but also directly the structural design (shape, number of barbs, etc.) of a feather. Therefore, feather growth-rate is severely constrained by the requirements imposed by the structural feather design.

## Introduction

Feathers assume many vital functions in birds (e.g. protective barrier, thermal insulation, flight). Full-grown feathers are dead structures and deteriorate with time. Therefore, feathers need to be periodically replaced, a process known as moult. However, feather replacement presents two major challenges.

**Competing interests:** The authors have declared
that no competing interests exist.

First, feathers can only regrow from a fixed number of feather follicles which have developed during embryonic life [1–4]. Second, feathers lack the capacity for self-repair and they also do not regrow continuously from a living basal tissue, like cornified structures such as claws and hairs. Therefore, feathers need to be replaced entirely, the newly developing feather pushing out the old one [5]. When a feather is dropped or growing, it cannot assume full functionality, and the regrowing feather is fragile and vulnerable. As a consequence, moult causes gaps in the plumage and a reduction in plumage functionality (e.g. impaired flight capability, reduced plumage insulation), depending on the type and number of simultaneously regrowing feathers. Therefore, a high feather growth-rate would be advantageous, because the faster a feather, and hence the plumage, is replaced, the shorter is the time of reduced plumage functionality.

However, it has long been known that feather growth-rate is limited (e.g. [6,7]). Growth-rates of the remiges (flight feathers) only differ by a factor of about 3 between small and large birds, while the length of the longest primary varies by a factor of more than 10. The length of the primary feathers (F) increases isometrically with body mass (M) across species (F ≈ $M^{0.33}$; [8–10]), while the growth-rate (G) of primaries increases only slightly with body mass (G ≈ $M^{0.171}$; [7]); therefore, a long primary needs disproportionally more time to grow than a small primary. As a consequence, large species cannot moult all feathers within the time available in the annual cycle if they need to preserve some degree of flight capability [7], and the limitation of feather growth-rate is a serious constraint, particularly for large birds.

Two main reasons have been offered to explain why feather growth-rate does not scale with feather-length: (a) limitation of the rate of cell division at the base of a feather, within the epidermal collar, i.e. limitation of growth-rate by the size of the epidermal collar; (b) limitation of nutrient supply to the epidermal collar and the growing feather.

Prevost [6] argued that "the cells producing feathers are basically the same, and are independent of the size of the bird"; and that "the rate of feather growth must at some point be limited by rates of cell division, and this limit may well be similar for all species". Also Prum & Williamson [11] in their model of feather growth consider the growth-rate at the epidermal collar (i.e. the cell division rate) a decisive factor for shaping feathers. Rohwer and colleagues [7,12] consider the growth zone of the epidermal collar a linear structure (in a circular configuration) which produces a two-dimensional feather, and posit that growth-rate should theoretically relate to feather-length with an exponent of 0.5 (which they found in their data) under the assumption of isometry. They suggest that the circumference of the epidermal collar limits feather growth-rate.

Feather growth may also be limited by nutrient supply to the epidermal collar, e.g. the rate at which proteins diffuse through the collar of cells surrounding a developing feather [13,14]. Furthermore, nutrient supply may limit feather growth in the ramogenic zone and further up, where differentiation, elongation and keratinization of the cells occurs, and these processes may depend on the supply of nutrients via the feather pulp, which extends into the growing epidermal cylinder (Fig 1).

Most attempts to explain feather growth-rate have used growth-rate by length, rather than growth-rate by mass. However, if the growth processes in the epidermal collar and ramogenic zone are limited by cell division and/or nutrient transport, growth-rate by mass may be physiologically more relevant, and hence a more appropriate measure of feather growth. Growth-rate by length might be a poor proxy of the rate of feather material produced, given the different shapes and structures (e.g. massiveness [mass of feather material per mm feather length]) of feathers between and within species. On the other hand, growth-rate by length is the important measure regarding the period of reduced feather functionality, and hence moult duration.

In this study, we first examine whether feather growth-rate depends on the size of the linear (ring) structure of the epidermal collar. The epidermal collar is at the base of the feather follicle

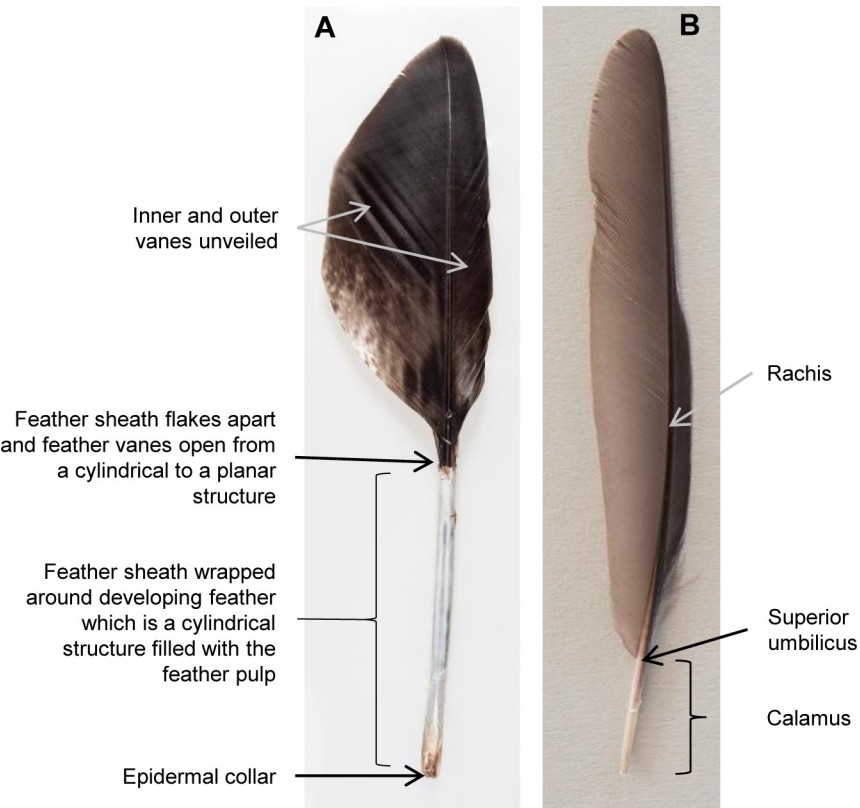

**Fig 1. Morphology of a growing feather.** (A) A growing feather (primary of a Golden Eagle *Aquila chrysaetos*). Essentially all feather growth takes place within the ring structure of the epidermal collar of the follicle which is located at the base of the feather [4]. The epidermal collar produces cells which migrate up and proliferate the cells which produce the feather keratinocytes arranged in the form of a cylinder [15–18]. Nutrient supply for the differentiation, elongation and keratinization of the keratinocytes is via the feather pulp, which fills the cylinder of the developing feather. When the feather pulp retreats and the feather sheath brakes away the vanes unfold. (B) Full-grown feather (primary of a Common Blackbird *Turdus merula*). When the production of barb ridges ends, the cylinder, which opens to unveil the feather vanes, turns into the calamus which maintains its cylindrical shape and therefore represents the diameter of the feather follicle during the last phase of growth (e.g. 4). Therefore, we measured the diameter of the calamus in the dorso-ventral and distal-proximal direction at the transition to the rachis (at the superior umbilicus, which is the widest point of calamus and rachis) as a proxy of epidermal collar size.

and follicle diameter is determined by the barb and rachis ridges it contains [4,19]. When the production of barb ridges ends, the cylinder, which opens to unveil the feather vanes (barbs attached to the rachis), turns into the calamus which maintains its cylindrical shape and therefore represents the diameter of the feather follicle during the last phase of growth (e.g. [4]; Fig 1). Second, we examine how follicle size, and consequently the rate of feather mass produced, relates to the length and structure (massiveness) of the feather. Third, these relationships are then used to explain the observed low increase of feather growth-rate with feather-length. Finally, we explain why the size of the feather follicle cannot be increased to allow for a higher feather growth-rate.

By using allometric relationships we analysed three independent datasets: (1) The longest primary wing-feather of 27 species. In this dataset, the type of feather is the same across all species, i.e. the primary forming the wing-tip. The longest primary takes the longest to regrow and, therefore, sets the theoretical minimum time of moult (i.e. when all feathers are moulted concurrently). (2) Different feather types (i.e. primaries and secondaries) of the Golden Eagle *Aquila chrysaetos* to see whether the growth of different feather types (differently shaped and

constructed) is limited by the same factors. (3) All primary wing-feathers of 6 passerine species; this is interesting because the primary forming the wing-tip and all primaries distal to it are heavier given their length than the proximal primaries [20], and so can be used to see how the growth of primaries of different mass varies.

## Materials and methods

### Feather data

The first dataset consisted of the longest primary of 27 species (S1 Table) ranging from a small 9 g passerine (Willow Warbler *Phylloscopus trochilus*) to one of the largest flying birds with a flapping flight, the Mute Swan *Cygnus olor* (10 kg). From 1–4 dead individuals per species, we plucked the longest primary of one wing and measured its overall mass and length. We measured the diameter of the calamus in the dorso-ventral and distal-proximal direction at the transition to the rachis (at the superior umbilicus, which is the widest point of calamus and rachis) as a proxy of epidermal collar size (Fig 1). We calculated the mean for species with more than one individual. Assuming an elliptical cross-section, we calculated the circumference and the cross-sectional area of the calamus, to see whether a linear or two-dimensional measure is directly proportional to feather growth-rate. Using growth-rates of the longest primary from the literature or from own data (S1 Table), we calculated mean growth-rates by length (mm per day) and mean growth-rates by mass (mg per day), as well as the feather material used on average to build 1mm of feather-length (called feather massiveness; mg feather material per mm feather-length, calculated by dividing feather mass by feather-length).

The second dataset consisted of 45 plucked flight-feathers (14 primaries and 31 secondaries) from 6 dead Golden Eagles (4–13 primaries and/or secondaries per individual). We determined the growth-rate by length of each feather from growth bars [21,22]. Growth bars are alternating light and dark bands across the rachis and vane; one pair of dark and light bands represents the growth increment during 24 h [23]. Other measures were taken or calculated as described above.

The third dataset consisted of all primaries of six passerine species. Growth-rates by length of each primary were taken from the literature: Eurasian Magpie *Pica pica* [24]; House Sparrow *Passer domesticus* ([25], read from Fig 3); White-winged Snowfinch *Montifringilla nivalis*, ([26], mean growth-rate up to total length of the post-juvenile feathers); European Greenfinch *Chloris chloris* and Eurasian Bullfinch *Pyrrhula pyrrhula* [27]; European Starling *Sturnus vulgaris* ([28], maximum growth-rates read from Fig 5). These species have 9 functional primaries, except the Eurasian Magpie which has 10. Of all primaries, overall mass and length, and the diameter in the dorso-ventral and distal-proximal direction at the superior umbilicus of the calamus were measured on feathers from one adult dead individual per species to calculate the parameters mentioned above. Because the growth-rates taken from the literature were determined with various methods (maximum growth-rate, or mean growth-rate of the entire feather, or of the feather emerging from the skin), the growth-rates and derived parameters cannot be compared directly between species. For each species we determined the primary forming the wing-tip from museum specimens (which agreed with that in [20]) and named this primary and all distal to it the 'wing-tip primaries', while the primaries proximal to the primary forming the wing tip are called 'proximal primaries'. The feathers of all datasets were intact with no or only little abrasion.

All feathers are from animals received after death to the Swiss Ornithological Institute or the Natural History Museum Basel, and no birds were killed for the purpose of this study. Hence no permits were required.

## Statistical analyses

We analysed the data within the framework of biological scaling or allometry [29,30], where the relationship between two traits x and y is expressed as $y = kx^a$. The exponent $a$ is called the 'scaling exponent' or 'allometric coefficient'. Such relationships are usually analysed in their logarithmic (to base 10) form $\log y = a \log x + \log k$ and plotted on a log-log scale. For estimating the scaling exponent usually reduced major axis regression is applied, because it accounts for the variation in both variables, contrary to ordinary least-squares regression which does not account for error variance in the independent variable.

Considering the dataset of the longest primary of 27 species, we ran Reduced Major Axis (RMA; [31]) regressions for a set of relationships between various feather traits (S2 Table) using the 'lmodel2' function available in the homonymous R package [32]. To account for non-independence between species owing to shared evolutionary history, we also applied Phylogenetic RMA (PhyloRMA) regressions for the same set of relationships using the 'phyl.RMA' function in the 'phytools' R package [33]. Specifically, we retrieved a complete phylogeny of our 27 species from the BirdTree database (http://birdtree.org; [34]), creating a total of 1000 trees based on 'Ericson' phylogenetic reconstruction ($n_{species}$ = 9993). For each of the 1000 trees we ran a PhyloRMA and then estimated the average and 95% confidence intervals for the intercept and slope of each of the relationships. For the majority of regressions, RMA and PhyloRMA produced nearly identical results (S2 Table), which indicated a very small effect of shared evolutionary history among our species on the feather traits measured. Therefore, we present only the results of RMA in the figures and Tables 1 and 3. We also tested whether the estimated RMA slopes were significantly different from a set value (representing isometry, i.e. the preservation of proportionality between traits across the range of their values) using the 'slope.test' function available in the 'smatr' R package [35]. Finally, we tested the combined effects of two feather traits on feather growth-rate by length using Generalized Linear Models (GLMs; Table 2), rather than RMA or PhyloRMA, because these do not allow for the simultaneous inclusion of more than one covariate.

For the remiges of Golden Eagles, we ran RMA analyses for various relationships between feather traits (S3 Table) and tested whether the resulting RMA slopes were significantly different from expected isometry. We tested whether 'feather type' (binary fixed term to distinguish between primaries and secondaries which have a different shape and structure) had a

**Table 1. Reduced major axis regressions between feather growth-rate by mass, and alternatively feather growth-rate by length, and the cross-sectional area of the calamus as a proxy of feather follicle size.**

|  | Intercept | 95% CI | Slope | 95% CI | $R^2$ |
|---|---|---|---|---|---|
| **Longest primary of 27 species** |  |  |  |  |  |
| log(growth-rate by mass) ~ log(calamus cross-sectional area) | 2.901 | 2.866, 2.935 | 1.142* | 1.078, 1.211 | 0.981 |
| log(growth-rate by length) ~ log(calamus cross-sectional area) | 0.523 | 0.491, 0.549 | 0.236 | 0.185, 0.297 | 0.755 |
| **Remiges of Golden Eagle** |  |  |  |  |  |
| log(growth-rate by mass) ~ log(calamus cross-sectional area) | 0.253 | 0.141, 0.356 | 0.971¨ | 0.898, 1.049 | 0.941 |
| log(growth-rate by length) ~ log(calamus cross-sectional area) | -0.590 | -0.726, -0.481 | 0.316 | 0.24, 0.411 | 0.572 |
| **Primaries of 6 passerine species** |  |  |  |  |  |
| log(growth-rate by mass) ~ log(calamus cross-sectional area) | -0.116 | -0.129, -0.104 | 1.046¨ | 0.929, 1.181 | 0.842 |
| log(growth-rate by length) ~ log(calamus cross-sectional area) | 0.471 | 0.443, 0.487 | 0.229 | 0.057, 0.487 | 0.096 |

Intercept and slope (scaling exponent) are given with their 95% confidence intervals (95% CI). For growth-rate by mass:

* = slope is significantly different (P < 0.001) from 1;

¨ = slope not significantly different from 1 (P > 0.32).

**Table 2. Dependence of the cross-sectional area of the calamus, or alternatively growth rate by mass, on both feather-length and feather massiveness (mass of feather material per mm feather-length).**

| | Intercept | 95% CI | log(feather-length) | | log(feather massiveness) | | $R^2$ |
|---|---|---|---|---|---|---|---|
| | | | Slope | 95% CI | Slope | 95% CI | |
| **Longest primary of 27 species** | | | | | | | |
| log(calamus cross-sectional area) | -0.826 | -1.749, 0.097 | 0.676 ** | 0.261, 1.090 | 0.723 *** | 0.505, 0.940 | 0.992 |
| log(growth-rate by mass) | 0.407 | -0.879, 1.693 | 0.121 ¨ | -0.456, 0.699 | 1.161 *** | 0.858, 1.464 | 0.988 |
| **Remiges of Golden Eagle** | | | | | | | |
| log(calamus cross-sectional area) | 1.629 | 0.537, 2.720 | 0.539 * | 0.134, 0.945 | 0.849 *** | 0.463, 1.235 | 0.94, 0.95 |
| log(growth-rate by mass) | 1.520 | 0.433, 2.607 | 0.650 ** | 0.246, 1.053 | 0.725 *** | 0.341, 1.110 | 0.94, 0.96 |
| **Primaries of 6 passerine species** | | | | | | | |
| log(calamus cross-sectional area) | -0.359 | -0.983, 0.266 | 0.473 *** | 0.200, 0.746 | 0.901 *** | 0.695, 1.106 | 0.98, 0.99 |
| log(growth-rate by mass) | -1.746 | -2.519, -0.973 | 1.099 *** | 0.758, 1.440 | 0.341 * | 0.062, 0.619 | 0.78, 0.99 |

Results from linear models for the longest primary of 27 species. Mixed effects models for the remiges of Golden Eagle and the primaries of six passerine species with marginal $R^2$ (without the random effect individual) and conditional $R^2$ (with the random effect individual). Significance of the slopes

*** P < 0.001

** P = 0.002 or 0.003

* P = 0.01 or 0.02,

¨slope not significantly different from 0.

significant effect with linear mixed models (LMM) with individual as a random effect (S4 Table). We also tested the combined effects of two feather traits on feather growth-rate by length using GLM and LMM (with individual as a random effect in the latter; S4 Table).

For the primaries of six passerine species, we ran both RMA and PhyloRMA analyses for various relationships between the feather traits. PhyloRMA and RMA analyses produced nearly identical results, so that we only present the results of RMA analysis (Tables 1 and 3, S5 Table). We also used LMM to test for an effect of 'wing-tip primaries' (binary fixed term) with

**Table 3. Reduced major axis regressions between feather growth-rate by mass and feather-length, between feather growth-rate by length and feather-length, as well as between feather mass and feather-length.**

| | Intercept | 95% CI | Slope | 95% CI | $R^2$ |
|---|---|---|---|---|---|
| **Longest primary of 27 species** | | | | | |
| log(growth-rate by mass) ~ log(feather-length) | -1.545 | -1.985, -1.139 | 2.344* | 2.155, 2.548 | 0.961 |
| log(growth-rate by length) ~ log(feather-length) | -0.392 | -0.673, -0.163 | 0.482¨ | 0.376, 0.613 | 0.744 |
| log(feather mass) ~ log(feather-length) | -4.222 | -4.535, -3.924 | 2.895¨ | 2.756, 3.041 | 0.986 |
| **Remiges of Golden Eagle** | | | | | |
| log(growth-rate by mass) ~ log(feather-length) | -0.600 | -0.811, -0.404 | 1.447* | 1.321, 1.584 | 0.920 |
| log(growth-rate by length) ~ log(feather-length) | -0.853 | -1.081, -0.671 | 0.462¨ | 0.343, 0.609 | 0.544 |
| log(feather mass) ~ log(feather-length) | -2.847 | -2.984, -2.715 | 2.050* | 1.965, 2.139 | 0.981 |
| **Primaries of 6 passerine species** | | | | | |
| log(growth-rate by mass) ~ log(feather-length) | -4.011 | -4.606, -3.499 | 2.075* | 1.811, 2.384 | 0.801 |
| log(growth-rate by length) ~ log(feather-length) | -0.329 | -1.245, 0.312 | 0.426* | 0.094, 0.901 | 0.093 |
| log(feather mass) ~ log(feather-length) | -3.721 | -3.981, -3.474 | 2.669* | 2.541, 2.804 | 0.969 |

* = slope is significantly different (P < 0.001) from 3 (in the case of growth-rate by mass against feather-length and feather mass against feather-length), or from 0.5 (in the case of growth-rate by length against feather-length);

¨ = slope not significantly different from 3 or 0.5, respectively (P > 0.90). For models including feather type of Golden Eagles and passerine primaries, see S4 Table and S6 Table.

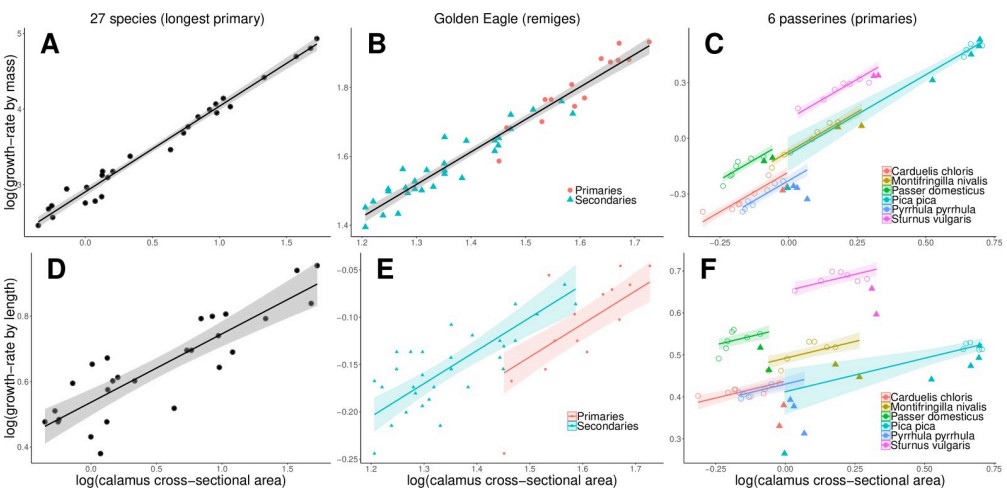

**Fig 2. Feather growth-rate and cross-sectional area of the calamus.** Relationships between feather growth-rates by mass (A—C), or alternatively feather growth-rate by length (D—F), and the cross-sectional area of the calamus for three datasets: A and D: longest primary of 27 species, B and E: remiges of Golden Eagle (dots = primaries, triangles = secondaries), C and F: primaries of 6 passerine species (colours; filled triangles = 'wing-tip primaries', open circles = 'proximal primaries'). The lines indicate the regressions according to the analyses (see Table 1 and S2 Table, S4 Table and S6 Table): A, B, D: for the entire dataset; E: with a different intercept for primaries and secondaries; C and F: for the 'proximal primaries' only with a different intercept according to species.

species as a random effect (S6 Table). Again, we tested the combined effects of two feather traits on feather growth-rate by length using GLM and LMM (with species as a random effect in the latter; S6 Table).

We deliberately provide a whole range of relationships in the S2–S6 Tables, to facilitate comparison with published or oncoming work; the most important relationships are presented in Tables 1–3.

## Results

### Scaling of growth-rates by mass with calamus width

In all three datasets, we found strong positive correlations between feather growth-rate by mass and the cross-sectional area of the calamus (Table 1, Fig 2A–2C), or alternatively its circumference (S2 Table, S3 Table and S5 Table). For the longest primary of the 27 species (Fig 2A), these correlations and their major axis slopes were nearly independent of the phylogenetic relationships of the species (S2 Table). In the case of the different flight-feathers of Golden Eagles, these correlations did not differ significantly between primaries and secondaries (S4 Table). For the primaries of the six passerine species, there was a significant difference between the 'proximal' and the 'wing-tip primaries' (S6 Table, Fig 2C). The 'wing-tip primaries' had a significantly lower growth-rate by mass than expected from their calamus width, compared with the proximal primaries. Furthermore, the intercept differed between species (S6 Table), probably due to differences in the way growth-rates were measured (see Materials and Methods).

In all three datasets, the scaling exponents of the relationships between feather growth-rate by mass and the circumference of the calamus were close to 2 (S2 Table, S3 Table and S5 Table), and with the cross-sectional area of the calamus close to 1 (Table 1). The latter indicates direct proportionality: a doubling in cross-sectional area of the calamus is linked to a doubling in growth-rate by mass. In detail, the major axis slope of the relationship between feather

growth-rate by mass and the cross-sectional area of the calamus for the 27 species was slightly but significantly higher than 1 (and slightly higher than 2 for the circumference of the calamus), while there was no significant difference from isometry in the Golden Eagle dataset and the primaries of the 6 passerine species (Table 1).

## Scaling of growth-rates by length with calamus width

In all three datasets, growth-rate by length was much less strongly correlated with the cross-sectional area of the calamus (Fig 2D–2F, Table 1; or alternatively with its circumference, S2 Table, S3 Table and S5 Table) than growth-rate by mass.

In the case of the remiges of Golden Eagles and the primaries of the six passerines, growth-rate by length differed significantly between the types of feathers in addition to being dependent on calamus diameter. Primaries of Golden Eagles had a significantly lower growth-rate by length than secondaries (effect size on the log-log scale -0.042 ± 0.015 SE, P < 0.001; see S4 Table), and the same was true for 'wing-tip primaries' compared with 'proximal primaries' in the six passerines (effect size on the log-log scale -0.072 ± 0.007, P < 0.001; see S6 Table).

## Scaling of calamus width with feather-length and feather structure

In all three datasets, the cross-sectional area of the calamus correlated strongly with both feather-length and feather massiveness (mass of feather material per mm feather-length; Table 2; see Fig 3 and S2 Table, S3 Table and S5 Table for the reduced major axis relationship only with feather-length). This means that the size of the feather follicle increases with feather-length and, for a given feather-length, additionally with the mass of feather material deposited per mm length. In fact, feather massiveness had a stronger effect on feather follicle size than feather-length (Table 2).

The type of feather ('wing-tip' *vs* 'proximal primaries') had no significant effect on calamus width if both feather-length and feather massiveness were included in the model (effect size 0.013; P = 0.292), although there was an effect in a model with only feather-length (S6 Table). This indicates that, within the primaries of passerines, the more massive structure of 'wing-tip primaries' (which have more feather mass deposited per mm feather-length, see S6 Table) is linked to a larger feather follicle. In Golden Eagles, the lengths of primaries and secondaries overlap only little, and the inner primaries are more like secondaries in structure, so that an analogous analysis would be inappropriate.

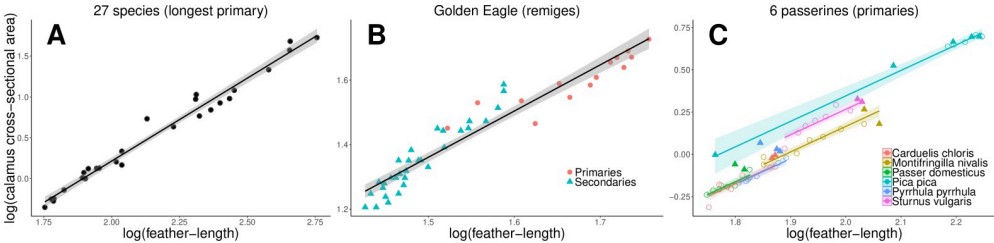

**Fig 3. Cross-sectional area of the calamus and feather-length.** Relationships between the cross-sectional area of the calamus and feather-length for three datasets: A: longest primary of 27 species, B: remiges of Golden Eagle (dots = primaries, triangles = secondaries), C: primaries of 6 passerine species (colours; filled triangles = 'wing-tip primaries', = open circles 'proximal primaries'). The cross-sectional area of the calamus in addition depends on the mass of feather material per mm feather-length which is not depicted (see text and Table 2). The lines indicate the regressions according to the analyses (see Table 2 and S2 Table, S4 Table and S6 Table): A and B: for the entire dataset; C: for the 'proximal primaries' only with a different intercept according to species.

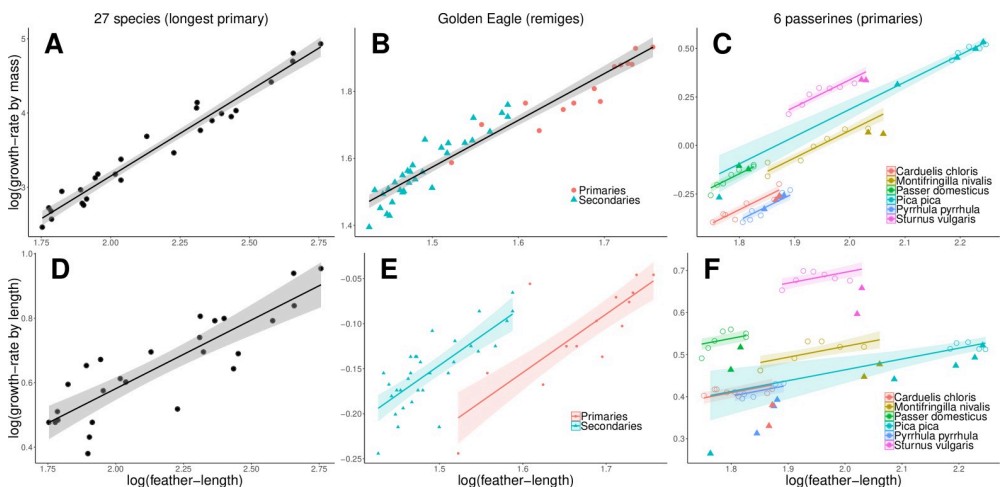

**Fig 4. Feather growth-rate and feather-length.** Relationships between feather growth-rates by mass (A-C), or alternatively growth-rate by length (D-F), and feather-length for three datasets: A and D: longest primary of 27 species, B and E: remiges of Golden Eagle (dots = primaries, triangles = secondaries), C and B: primaries of 6 passerine species (colours; filled triangles = 'wing-tip primaries', open circles = 'proximal primaries'). The lines indicate the regressions according to the analyses (see Table 2 and S2 Table, S4 Table and S6 Table): A, B, D: for the entire dataset; E: with a different intercept for primaries and secondaries; C and F: for the 'proximal primaries' only with a different intercept according to species. Golden Eagle growth-rate by length: Note that the outlier primary near the upper end of the regression line for secondaries is the only innermost primary analysed which is more similar in shape to the secondaries than the other primaries.

## Scaling of growth-rates by mass with feather-length

Because growth-rate by mass correlated strongly with the cross-sectional area of the calamus, and the cross-sectional area of the calamus with both feather-length and the massiveness of the feather, there was also a strong correlation between growth-rate by mass and both feather-length and feather massiveness (Table 2; for the reduced major axis relationships only with feather-length see Fig 4A–4C and S2 Table, S3 Table and S5 Table). As with follicle size, this means that growth-rate by mass increases with feather-length and, for a given feather-length, additionally with the mass of feather material deposited per mm length.

The type of feather ('wing-tip' vs. 'proximal primaries') had a significant effect on growth-rate by mass in a model with feather length and mass per length as independent variables, and species as random effect (S6 Table). The 'wing-tip primaries' had a slightly lower growth-rate by mass than expected from their length and massiveness (effect size 0.057 ± 0.007 SE). In Golden Eagles, the length of primaries and secondaries overlap only little, so that, as above, an analogous analysis would be inappropriate.

The scaling exponents of growth-rate by mass against feather-length (2.34 for the 27-species dataset, 1.45 for the Golden Eagle remiges, and 2.08 for the passerine primaries) were all significantly different from 3, which is the isometric exponent, and lower than the respective scaling exponents of feather mass against feather-length (Table 3).

## Scaling of growth-rates by length with feather-length

In all three datasets, the correlations between growth-rate by length and feather-length were much lower, compared with the analogous correlations with growth-rate by mass (Table 3, cf. Fig 4D, 4E and 4F with Fig 4A, 4B and 4C). In the Golden Eagle dataset, there was a significant effect of feather type (effect size 0.073 ± 0.016 SE, P<0.001), and including feather type into the model changed the slope to 0.649 ± 0.078 (S4 Table). For a given size, primaries had a

lower growth-rate by length than secondaries (Fig 4E), because primaries are more massive (see S4 Table). Within the primaries of the six passerines species, the 'wing-tip primaries' had a lower growth-rate by length than the 'proximal primaries' (effect size 0.063 ± 0.006, P<0.001, Fig 4F), and including feather type into the model changed the slope to 0.377 ± 0.041 (S6 Table).

The scaling exponents of growth-rate by length against feather-length derived from the three datasets were between 0.43 and 0.48 when not accounting for feather type (Table 3), hence slightly lower than 0.5 which is the isometric exponent. However, if accounting for feather type, the exponents varied between 0.38 and 0.65 (S4 Table and S6 Table).

## Discussion

This study revealed three main findings: (1) The cross-sectional area of the calamus, as a proxy of feather follicle size, correlated strongly with feather growth-rate by mass across species, and across feather types (primaries and secondaries in Golden Eagles, primaries of 6 passerine species); a doubling in cross-sectional area of the calamus entailed a doubling in growth-rate by mass. (2) The cross-sectional area of the calamus (follicle size) was related to both feather-length and feather structure (feather massiveness): longer and more massive feathers (more material per mm length) had a larger feather follicle. (3) Growth-rate by mass correlated well with feather-length with scaling exponents between 1.45 and 2.34, which are lower than the scaling exponents of feather mass against feather length. Growth-rate by length did not correlate as well with feather-length, because feather massiveness had a large additional effect.

The results provided here for the longest primary of 27 species may hold for birds in general, because we covered a large range of species across 7 bird orders and 16 families with a large range in primary feather-length (55–571 mm). Moreover, accounting, or not, of the phylogenetic relationships between species barely altered the results (S2 Table).

### Feather mass production depends on follicle size

We found that feather growth-rate by mass was tightly correlated with the width of the calamus. For reasons given in the introduction, we assume that calamus diameter at the superior umbilicus (the widest point) represents follicle diameter and hence the width of the epidermal collar at the time when this part of the feather had been growing.

The 'wing-tip primaries' of passerines had a relatively lower growth-rate by mass than expected from their calamus width, compared with the proximal primaries (Fig 2C). This can be explained by the finding of Dawson [28]: the bottom part of the 'wing-tip primaries' is comparatively heavier than in the 'proximal primaries', while the upper feather part is similar or lighter in massiveness to the 'proximal primaries'. Because we measured calamus diameter at the superior umbilicus, it represents the follicle size for growing the bottom part of the feathers. Therefore, calamus width at the bottom part of the 'wing-tip primaries' overestimates overall growth-rate by mass which we averaged over the entire feather.

The cross-sectional area of the calamus, rather than its circumference, was directly proportional to feather growth-rate by mass. This may indicate that it is not the linear arrangement of stem cells (in a circular configuration) which primarily determines growth-rate, but some parameter acting at a two-dimensional scale, such as nutrient supply. Stem cells and their derived cells in the epidermal collar can only divide when supplied with nutrients from the surroundings and similarly, the subsequent development of the keratinocytes (elongation, differentiation and keratinization) depends on nutrients supplied by the pulp. These supplies are probably specific to the stage of feather development along the feather axis and, therefore, may

be proportional to the cross-section of the developing feather. However, further study is required to reveal the precise physiological limitations of feather growth.

## Follicle size is adapted to feather structure

Follicle size was positively related to both feather-length and feather massiveness (feather mass per mm feather-length). For example, the 'wing-tip primaries', by being more massive for their size, had a proportionally larger follicle size than the less massive proximal primaries, as also found for the outermost primary of the Grey Plover *Pluvialis squatarola* [36]. This indicates that follicle size is adapted to feather structure (both length and massiveness). A feather of a given length has a slightly larger follicle diameter, and hence a slightly higher growth-rate by mass, when it is to become a massive feather compared with a light feather.

As mentioned above, we measured a proxy of follicle size (calamus width at the superior umbilicus) when growing the bottom part of the feathers. The question is whether follicle size is adapted during the growth of a feather, in particular when growing the light tip of the feather. Growth-rates by mass of single primaries have been found to be nearly constant, and to level off only at the very end of growth [28,24], and this would suggest that follicle size also remains constant. However, growth of the light feather tip occurs predominantly in the skin or when the feather quill is just emerging, and is therefore generally missed by conventional growth measurements. It would be interesting to study in more detail the variation in follicle size during feather growth. For example, in Indian Peacock *Pavo cristatus* train feathers the feather eyes not only consist of much more material deposited per mm length, and grow at a slower rate, but also grow with a larger diameter of the feather follicle compared with the growth of the loosely barbed remainder of the feather which grows quickly and from a follicle shrunk in diameter [37].

## Follicle size and feather structure determine growth-rate by length

Given that follicle size (the cross-sectional area of the calamus) determines the rate of feather mass produced, it depends on the particular feather on how this mass is distributed within the feather, i.e. whether a massive feather with a lot of feather mass per length, or a light feather with less material per length, is produced. And, given a certain rate of feather mass production, this deposition rate of mass per unit length in turn determines the growth-rate by length. Indeed, we found that feather growth-rate by length is determined by both the cross-sectional area of the calamus (which determines the rate of feather mass produced) and feather type in Golden Eagles remiges and passerine primaries (see Fig 5). The more massive primaries of Golden Eagles and 'wing-tip primaries' of passerines had a lower growth-rate by length than the secondaries and proximal primaries, respectively, after accounting for feather length.

## Scaling of growth-rates with feather-length and body size

The scaling exponents of feather mass against feather length were close to, but slightly lower than 3 (isometry) when examined across species, and around 2.1–2.7 within species (Table 3), which agrees with literature data. Thus, in a dataset of 120 species, mean primary feather mass (calculated across all primaries of a species) scales with mean primary feather-length with an exponent of 3.0, while within species exponents are generally lower (around 2.4), but vary between species and their habitats, and reflect how primary feather mass is distributed among the primaries [20]. For the outermost rectrix of 98 Nearctic species, the scaling exponent of feather mass against feather-length is 2.97 [38]. Hence, primaries of large birds are slightly shorter, and also less massive and more flexible than those of small birds [9,10].

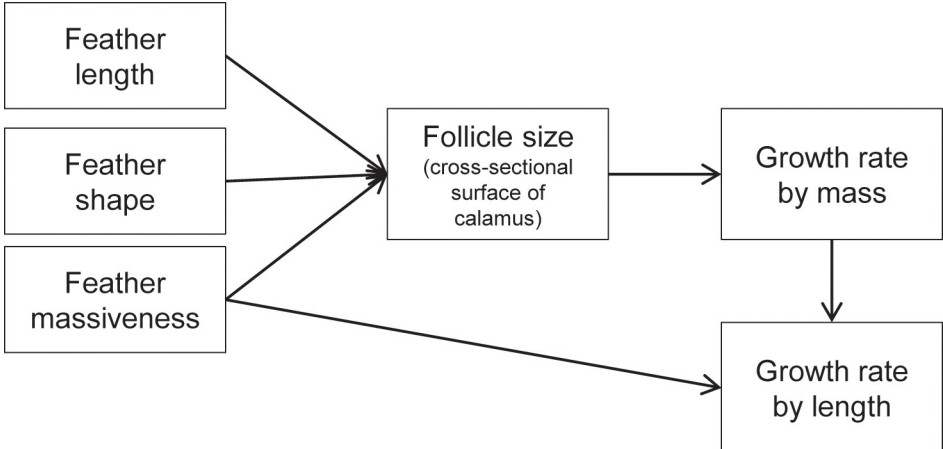

**Fig 5. Schematic representation of the determinants and constraints of feather growth rates, as suggested by this study.**

Despite large feathers being proportionally slightly less massive, the scaling exponents of growth-rate by mass against feather-length were much lower than the scaling exponents of feather mass against feather length in all three datasets (Table 3). This clearly shows that feather growth by mass of large feathers does not keep up with that of small feathers, and large feathers take disproportionally longer to grow. Taking the relationships of the 27 species as an example, a doubling of primary length entailed a 7.4 times higher feather mass, but feather mass produced per day increased by only 5.1 times, which results in a 1.5 times longer primary-growth duration.

Feather growth by mass of large feathers does not keep up with that of small feathers because follicle size (cross-sectional area of the calamus) increased with feather-length with scaling exponents of 2.05, 1.45, and 1.98 (for the three datasets, respectively; see S2 Table, S3 Table and S5 Table; not accounting for the effect of massiveness), while feather mass increased with feather-length with much higher exponents (Table 3). We found only two studies which measured rachis diameter over a range of species: Worcester [8] calculated a scaling exponent of 1.16–1.19 of the dorso-ventral diameter of the longest primary at the skin area across 13 species; Wang & Clarke [39] found a scaling exponent of 0.94 across 73 species (no indication where exactly and in which direction rachis diameter was measured).

Because growth-rate by length depended on both follicle size (which determines feather mass production rate) and the massiveness of the feather, the correlation between feather-length and growth-rate by length became poorer the more similar feathers were in length. Across a large range of species (and hence also primary feather-lengths), the correlation was still significant (Fig 4D). In contrast, remiges with lengths of the same order of magnitude had very similar growth-rates by length, which depended more on feather massiveness than feather-length (Fig 4E and 4F). Within the primaries of the 6 passerine species, there is a particularly low correlation between growth by length and feather-length, mainly because growth-rates by length of the 'wing-tip primaries' are lower than those of the 'proximal primaries'. A slower growth of outer than inner primaries has been observed by several authors [e.g. 40,41,27,26,42,28]. Therefore, scaling exponents for the relation between growth-rate by length and feather-length, without taking into account the massiveness of the feathers, are only useful across a large range of feather-lengths. Across the 27 species, we found an exponent for the relation between growth-rate by length and feather-length of 0.48 which is similar to those

found by others: 0.47 for primaries, 0.46 for secondaries, 0.59 for rectrices and 0.53 for greater coverts of 27 species [12], 0.5 for primaries of 43 species ([7]; data taken from the literature with often imprecise growth-rates), 0.74 for the outermost rectrix of 98 Nearctic species [38]. Therefore, across species, growth-rate by length does not keep up with increasing primary length. Taking the relationships of the 27 species again as an example, a doubling of primary length entails only a 1.4 times higher growth-rate by length, which results in a 1.43 times longer primary growth duration.

Several authors have compared feather growth-rates by length with body size (usually taking body mass as a proxy). Scaling feather growth-rates with body size introduces additional variation, because the length and structure of primaries of a bird of a given size depends on flight style, and flight style on body size [20,43,44], and again is only meaningful across a large range of body size. Feather-length of primaries scales with body mass isometrically with an exponent of about 0.33, i.e. approximately isometrically (0.32, [8]; 0.325 and 0.316 for the longest primary and 0.316 for the sum of all primaries, [7], see also [45]; 0.30, [10]; 0.355 this study, see S2 Table). Combined with the exponent of 0.48 for the relation growth-rate by length against feather-length, this would give an exponent of 0.158 for the relationship feather growth-rate by length against body mass, which is close to the exponent of 0.171 found among the 27 species in our sample, and 0.171 found by Rohwer and colleagues [7,12]. This confirms that the primaries of birds of large body size take disproportionally more time to grow than the primaries of small-bodied birds [7,6]. Note that the scaling exponents are not entirely comparable between studies, because some authors used reduced major axis and others least-squares regressions which may give different results.

## Conclusions

Feathers grow day and night with a constant production of material [24,28,46]. The size of the feather follicle (cross-sectional area of calamus as a proxy) seems to determine the amount of feather material produced per time which then is used to construct a feather of a particular massiveness (mass per mm feather length). The size of the feather follicle is adapted to both the length and massiveness of the feather. Feather growth-rate by length is dependent on both the feather material produced per time (growth-rate by mass) and the amount of material deposited per unit feather-length (massiveness) (Fig 5). Because the size of the feather follicle (and hence feather mass produced per time) does not increase in direct proportion with total feather mass or feather length, but at a much lower rate, large feathers need disproportionally more time to grow than small feathers. As a consequence, this imposes time constraints on large birds when growing feathers: (a) large species cannot moult all feathers within the time available in the annual cycle if they need to preserve some degree of flight capability [7]; (b) nestlings of large species need a long time to grow their first set of flight-feathers which may prolong nestling time.

The question is why follicle size cannot be increased in large feathers so as to meet the increased feather mass production requirements and assure a faster feather growth-rate?

The reason is that follicle size not only determines the rate of feather mass production, but also the structural design of a feather (e.g. shape, number of barbs) and hence its physical properties (e.g. bending stiffness) (Fig 5). Follicle diameter is determined by the barb and rachis ridges it contains, and later, when the calamus is produced, by the diameter of the calamus [4,19]. In particular, the number of concurrently growing barb ridges is directly related to the circumference of the follicle [19]. Therefore, follicle diameter is one of the main factors which determines the structural design of a feather (see [11] for a model of feather shape and the various factors related to follicle size). Therefore, follicle size cannot be varied independently of

the structural design of the feather to suit requirements of a high growth-rate. As a consequence, feather growth-rate is severely constrained by feather structure.

## Supporting information

**S1 Table. Measurements of the longest primary of 27 species.**
(PDF)

**S2 Table. Reduced major axis (RMA) and phylogenetic RMA (PhyloRMA) regression of various parameters of the longest primary of 27 species.**
(PDF)

**S3 Table. Reduced major axis regressions of various parameters of the remiges (primaries and secondaries of Golden Eagles.**
(PDF)

**S4 Table. Linear mixed effects models applied to the parameters of Golden Eagle remiges.**
(PDF)

**S5 Table. Reduced major axis regressions of various parameters of the primaries of 6 passerine species.**
(PDF)

**S6 Table. Linear mixed effects models applied to the parameters of the primaries of 6 passerine species.**
(PDF)

## Acknowledgments

We thank David Jenny for providing the Golden Eagles, Raffaela Schmid for dissecting and measuring the primaries of the six passerine species, Gilberto Pasinelli for helpful comments and Robert A. Robinson for revising and improving the English.

## Author Contributions

**Conceptualization:** Lukas Jenni, Kathrin Ganz.

**Data curation:** Lukas Jenni, Kathrin Ganz, Pietro Milanesi, Raffael Winkler.

**Formal analysis:** Lukas Jenni, Pietro Milanesi.

**Methodology:** Lukas Jenni, Pietro Milanesi, Raffael Winkler.

**Project administration:** Lukas Jenni.

**Resources:** Kathrin Ganz.

**Supervision:** Lukas Jenni.

**Visualization:** Pietro Milanesi.

**Writing – original draft:** Lukas Jenni.

**Writing – review & editing:** Kathrin Ganz, Pietro Milanesi, Raffael Winkler.

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
