## [Decision Letter · Decision Letter 0]

13 Jan 2020

PONE-D-19-27977

Determinants and constraints of feather growth

PLOS ONE

Dear Dr. Jenni,

Thank you for submitting your manuscript to PLOS ONE. After careful consideration, we feel that it has merit but does not fully meet PLOS ONE’s publication criteria as it currently stands. Therefore, we invite you to submit a revised version of the manuscript that addresses the points raised during the review process.

We would appreciate receiving your revised manuscript by Feb 27 2020 11:59PM. To enhance the reproducibility of your results, we recommend that if applicable you deposit your laboratory protocols in protocols.io, where a protocol can be assigned its own identifier (DOI) such that it can be cited independently in the future. For instructions see: http://journals.plos.org/plosone/s/submission-guidelines#loc-laboratory-protocols

We look forward to receiving your revised manuscript.

Kind regards,

Petra Quillfeldt

Academic Editor

PLOS ONE

Journal Requirements:

Reviewers' comments:

Reviewer's Responses to Questions

**Comments to the Author**

1. Is the manuscript technically sound, and do the data support the conclusions?

Reviewer #1: Yes

Reviewer #2: Yes

2. Has the statistical analysis been performed appropriately and rigorously? 

Reviewer #1: Yes

Reviewer #2: Yes

3. Have the authors made all data underlying the findings in their manuscript fully available?

Reviewer #1: Yes

Reviewer #2: Yes

4. Is the manuscript presented in an intelligible fashion and written in standard English?

Reviewer #1: Yes

Reviewer #2: Yes

5. Review Comments to the Author

Reviewer #1: Though I have used feather growth rate as a dependent variable to study life history questions, I have not considered the physiology behind feather growth in the depth presented in this paper. The great strength of the paper is how they examined hypotheses about the control of feather growth from three perspectives. They tested feather growth concepts across species by holding feather ID constant and in choosing the longest primary as the focal feather, they chose the feather whose growth may be the rate limiting step in moult. They also held species constant (golden eagle) and looked at growth across feather types. Finally, they varied feather size while holding feather type constant (all the primaries) to look at what controls growth rate. I was happy with their statistical choices which took into account the non-independence of closely related species. In fact, their explanation of statistical procedures were worthy of a stats textbook.

I have no real issues with regard to publication. It may be, in part, because this is not my field. A minor complaint is that in the cross-species analysis there were a few species represented by only one individual. At this point, that is not worth pursuing. This paper would be of (slightly) wider interest if the authors discussed their results relative to the utility of using growth bars (mm/d) as an index of feather growth vs. mass/d. Everyone who uses ptilochronology as a technique would want to read this paper.

The study well-conceived and executed. It was a creative and thorough test of their hypotheses. The acknowledgements thanked “Robert A. Robinson for revising and improving the English.” He did a good job.

Reviewer #2: In this study, Jenni et al. examine the factors that influence feather growth in birds. By using three separate datasets, the authors clearly demonstrate that follicle size influences feather growth rate by mass, as well as structural design. In other words, the feather follicle size, which the authors estimate by measuring the cross-sectional size of the feather calamus, determines the amount of feather produced during a unit of time, then used in the production of a feather, which varies in massiveness (mass/length). These results help explain the observation that larger feathers need disproportionately longer to grow than small feathers.

Though growth rates in this study are based on estimates from the literature, as well as direct measurements, the results are robust. The use of three distinct datasets allows for thorough examination of within-individual, intraspecific, and interspecific variation. The analyses appear sound and I was happy to see that the authors included phylogenetic regressions and that those results seem to indicate that phylogenetic effects on these relationships are minimal.

Overall my comments are relatively minor. There are some issues with wording and clarity throughout. I’ve attempted to highlight a number of these spots. In addition, though the conclusions section was clear and well-written, the manuscript would benefit from additional text on implications.

Line 36: “two-dimensional parameter” isn’t clear to the reader and requires explanation or re-wording.

Line 37: “linear (circular)” reads as an oxymoron. Though this description is made clear later on in the text, it’s confusing in the abstract.

Line 38: missing word in “, hence seems”

Line 52: cut “for many reasons”

Line 57: suggest rewording to “, like cornified structures such as claws and hair”

Line 59: logic disagreement with “dropped” and the rest of the sentence. A dropped feather cannot be fragile or vulnerable

Line 66. Cite the first sentence.

Line 71: change “a little” to “slightly”

Line 89: cut “so”

Line 94: “ramogenic” should be defined.

Line 119-120: cut sentence.

Line 121-122: I’m not sure it makes sense to define a theoretical minimum time of moult if all feathers are never moulted concurrently.

Line 270: “although it had in a” – awkward wording.

Line 328-329: award wording, suggest rephrasing.

Line 347: two-dimensional scale still isn’t made clear to the reader.

Line 423: suggest different word than “acceptable”

Line 438: cut or reword “by far”

Fig. 5: Probably not necessary. If kept, needs to be modified.

6. PLOS authors have the option to publish the peer review history of their article (what does this mean?). If published, this will include your full peer review and any attached files.

Reviewer #1: No

Reviewer #2: No

---

## [Author Response · Author response to Decision Letter 0]

23 Mar 2020

We hope that we met all the style requirements.

On 24 March, all data will be made available on our public Institut's repository. We would appreciate to obtain the DOI of the PlosOne paper by then to refer to, if possible.

Done

---

## [Decision Letter · Decision Letter 1]

6 Apr 2020

Determinants and constraints of feather growth

PONE-D-19-27977R1

Dear Dr. Jenni,

We are pleased to inform you that your manuscript has been judged scientifically suitable for publication and will be formally accepted for publication once it complies with all outstanding technical requirements.

With kind regards,

Petra Quillfeldt

Academic Editor

PLOS ONE

Additional Editor Comments (optional):

Reviewers' comments:

Reviewer's Responses to Questions

**Comments to the Author**

1. If the authors have adequately addressed your comments raised in a previous round of review and you feel that this manuscript is now acceptable for publication, you may indicate that here to bypass the “Comments to the Author” section, enter your conflict of interest statement in the “Confidential to Editor” section, and submit your "Accept" recommendation.

Reviewer #2: All comments have been addressed

2. Is the manuscript technically sound, and do the data support the conclusions?

Reviewer #2: Yes

3. Has the statistical analysis been performed appropriately and rigorously? 

Reviewer #2: Yes

4. Have the authors made all data underlying the findings in their manuscript fully available?

Reviewer #2: Yes

5. Is the manuscript presented in an intelligible fashion and written in standard English?

Reviewer #2: Yes

6. Review Comments to the Author

Reviewer #2: The authors have responded to all major concerns in the review. Most comments were relatively minor and required for clarification and all have now been attended to.

7. PLOS authors have the option to publish the peer review history of their article (what does this mean?). If published, this will include your full peer review and any attached files.

Reviewer #2: No

---

## [Editor Report · Acceptance letter]

10 Apr 2020

PONE-D-19-27977R1 

Determinants and constraints of feather growth 

Dear Dr. Jenni:

I am pleased to inform you that your manuscript has been deemed suitable for publication in PLOS ONE. Congratulations! Your manuscript is now with our production department. 

With kind regards,

on behalf of

Dr Petra Quillfeldt 

Academic Editor

PLOS ONE